# Role of RBMS3 Novel Potential Regulator of the EMT Phenomenon in Physiological and Pathological Processes

**DOI:** 10.3390/ijms231810875

**Published:** 2022-09-17

**Authors:** Tomasz Górnicki, Jakub Lambrinow, Monika Mrozowska, Marzena Podhorska-Okołów, Piotr Dzięgiel, Jędrzej Grzegrzółka

**Affiliations:** 1Faculty of Medicine, Wroclaw Medical University, 50-368 Wroclaw, Poland; 2Division of Histology and Embryology, Department of Human Morphology and Embryology, Wroclaw Medical University, 50-368 Wroclaw, Poland; 3Division of Ultrastructure Research, Wroclaw Medical University, 50-368 Wroclaw, Poland

**Keywords:** target discovery, epithelial–mesenchymal transition (EMT), RNA-binding protein 3 (RBMS3), carcinogenesis, target therapy

## Abstract

RNA-binding protein 3 (RBMS3) plays a significant role in embryonic development and the pathogenesis of many diseases, especially cancer initiation and progression. The multiple roles of RBMS3 are conditioned by its numerous alternative expression products. It has been proven that the main form of RBMS3 influences the regulation of microRNA expression or stabilization. The absence of RBMS3 activates the Wnt/β-catenin pathway. The expression of c-Myc, another target of the Wnt/β-catenin pathway, is correlated with the RBMS3 expression. Numerous studies have focused solely on the interaction of RBMS3 with the epithelial–mesenchymal transition (EMT) protein machinery. EMT plays a vital role in cancer progression, in which RBMS3 is a new potential regulator. It is also significant that RBMS3 may act as a prognostic factor of overall survival (OS) in different types of cancer. This review presents the current state of knowledge about the role of RBMS3 in physiological and pathological processes, with particular emphasis on carcinogenesis. The molecular mechanisms underlying the role of RBMS3 are not fully understood; hence, a broader explanation and understanding is still needed.

## 1. Introduction

RNA-binding motif single-stranded-interacting protein 3 (RBMS3) is a glycine-rich protein that was described for the first time by D. Penkov et al. in 2000 [1]. The gene encoding this protein is called *RBMS3*, and it is located in the short arm of chromosome 3, specifically in the 3p24.1 region. The discovery of RBMS3 was an effect of the screening of human fibroblast cDNA with an upstream element of the a2(I) collagen promoter Box 5A. It belongs to the family of *c-Myc* gene single-strand binding proteins (MSSPs) involved in DNA replication, transcription, apoptosis induction, and cell–cycle progression [2,3]. Published papers provide evidence of the wide range of processes in which RBMS3 takes part in, including regulation of embryogenesis, pathogenesis of liver fibrosis, and bisphosphonate-related osteonecrosis of the jaw (BRONJ) [4,5,6]. From 2008 onwards, RBMS3 has become a potential prognostic marker of different types of cancer and a factor regulating the process of carcinogenesis [7,8]. In addition, recent articles have provided evidence of RBMS3 taking part in the epithelial–mesenchymal transition (EMT), a key process responsible for the creation of distant metastases [9]. 

RBMS3’s ability to suppress the growth and progression of different types of cancers makes it an interesting potential target for the development of novel anticancer therapies. There is still a need for a summary of the role of RBMS3 in physiology and pathology that would provide a synthetic evaluation of the information available about it. In this article, we are going to review and systematize the current state of knowledge about RBMS3 and its function in physiology and pathology, with a particular focus on its role in EMT.

## 2. Methods

The authors searched for topic-related materials in four big medical databases—PubMed, Embase, Ovid, and Scopus—on 12 December 2021. The searched keywords included: RBMS3, RNA-binding motif single-stranded-interacting protein 3, rbms3 cancer, rbms3 EMT, EMT, and epithelial–mesenchymal transition. The keywords were the same for all databases. The articles were screened for relevance and analyzed based on inclusion criteria. An article was presumed relevant if RBMS3 was directly mentioned by the authors of the research. References from all relevant articles were also reviewed to ensure the inclusion of all articles directly related to the topic of RBMS3 (Figure 1).

## 3. Role in Development and Physiology

It is stated that the expression of RBMS3 may be a part of the regulatory mechanisms of pancreas embryonic development in mouse models. Authors have shown a restricted expression of protein in the embryonic pancreas, the neural tube, and the dorsal root ganglion, with peak expression in the pancreas taking place at E13.5. RBMS3 acts on a post-transcriptional level and is able to bind to the 3′-UTR of pancreas transcription factor 1, the alpha subunit (*Ptf1α*) mRNA, stabilizing it and increasing the level of Ptf1α protein in cells. Ptf1α is responsible for exocrine cell differentiation. Scientists have also provided evidence of RBMS3 expression affecting the expression of various digestive enzymes and its role in maintaining the function of mature exocrine cells in the pancreas [4].

Experiments conducted on the zebrafish model brought to light the potential impact of RBMS3 on craniofacial development and chondrogenesis [10]. RBMS3 was discovered to be expressed transiently in the cranial neural crest, and its knockdown results in severe craniofacial defects. The authors point to the TGF-β receptor pathway as the mechanism responsible for these abnormalities. RBMS3 binds to and stabilizes the transcripts of the Smad2 pathway. Further studies discovered that RBMS3 also has the ability to interact with *Smad1*, as well as cell cycle regulators, such as the TGF-β, receptor *cyclin D1* and *Rac1* transcripts, introducing RBMS3 as a global regulator of chondrogenesis [10].

RBMS3 was also discovered to take part in preventing the degeneration of the nucleus pulposus. The mechanism underlying this process consists of decreasing the activity of the Wnt/β-catenin signaling pathway and targets, such as metalloproteinase-13 (MMP13) [11]. The study also showed that RBMS3 increases nucleus pulposus cell proliferation and decreases apoptosis, inflammation, and extracellular matrix degradation levels [11].

## 4. RBMS3 in Pathological Noncancerous Processes

From the moment of the discovery of RBMS3, scientists pointed to the role of this protein and the gene that encodes it in a wide range of pathological processes, including liver fibrosis, osteonecrosis of the jaw (ONJ), and exfoliation syndrome [12,13,14]. 

Liver fibrosis is a wound-healing type of EMT process serving as a response to the injury. It can potentially develop into cirrhosis and lead to organ failure as a consequence [12]. The focal point of this process lies in the activation of hepatic stellate cells (HSCs), which are responsible for the storage of vitamin A during their quiescent state but produce an excessive amount of the extracellular matrix after activation. One of the factors involved in inducing the activation of HSCs is the pair-related homeobox transcription factor Prx1, also involved in the production of collagen type α1(I). In their work, Fritz and Stefanovic provided evidence of the role RBMS3 has in the regulation of Prx1 expression. By binding to the 3’-UTR of the *Prx1* mRNA, RBMS3 stabilizes the structure of the mRNA, increasing the effectiveness of translation and the level of the Prx1 protein, thus leading to the stimulation of collagen type α1(I) gene transcription in HSCs. These results together with the post-transcriptional regulation of collagen type α1(I) expressions show a probable mechanism of RBMS3’s role in the onset of liver fibrosis [5].

Another domain of RBMS3 influence is its impact on bone density. There is evidence of a statistically significant interaction between the *RBMS3* and *ZNF516* genes that negatively impacts the hip bone mineral density (BMD). The study was conducted using the novel approach of genome-wide association studies (GWAS), a method that successfully unveiled a number of genetic loci that impact BMD [15,16]. Table 1 presents all the discovered single-nucleotide polymorphisms (SNPs) of the *RBMS3* gene discussed in this article. Although molecular mechanisms underlying this interaction are currently unknown, RBMS3’s impact on collagen expression may influence the extracellular matrix of bone tissue.

Another pathological process that may, among others, involve alteration in collagen type α1(I) expression is osteonecrosis of the jaw (ONJ). It is a serious adverse effect mainly connected to the administration of bisphosphonates (BPs), which are antiosteoclastic drugs used, among others, in oncological therapy to control bone metastasis and hypercalcemia. The frequency of ONJ ranges from 0.6% in breast cancer to even 15% in multiple myeloma [13]. Research conducted with the help of GWAS discovered a relation between the variation in the *RBMS3* gene and 5.8 times higher probability of developing bisphosphonate-related osteonecrosis of the jaw (BRONJ) [17]. Even though other researchers were not able to confirm this relation [13], taking into consideration the impact of RBMS3 on bone density postulated in [15], there is a wide area for researchers to establish the exact role of RBMS3 in ONJ [18].

Exfoliation syndrome (XFS) is an age-related systemic disease that is the most common risk factor for open-angle glaucoma, which can cause irreversible blindness. Based on familial aggregation studies, XFS is suspected to be a genetic disease. Specific loci in the *RBMS3* gene are proven to be correlated with susceptibility to XFS and exfoliation glaucoma, although the exact mechanism of this impact is yet to be discovered [14,19,25].

RBMS3 was also found to be potentially involved in autoimmune diseases. Specifically, there is evidence that certain SNPs in the *RBMS3* gene are responsible for an increased susceptibility to systemic sclerosis (SSc) and primary Sjögren’s syndrome (PSS) [20,21]. A weak correlation was also found between RBMS3 and periodontal disease [22].

The versatility of RBMS3 reaches even the field of psychiatric health care and neurodegenerative diseases, since various authors have linked it to resistance to antidepressant therapy and susceptibility to schizophrenia and amyotrophic lateral sclerosis (ALS) [26,27,28]. Gastrointestinal dysfunction is a common symptom in the autism spectrum disorder (ASD). The exact underlying mechanism of this process is unknown, but researchers revealed that in a specific group of patients with *FOXP1* haploinsufficiency, downstream targets of the Foxp1 protein are dysregulated in the mice model. One of these targets is RBMS3, thus providing additional data about its role in this disorder [29].

RBMS3’s impact seems to not be restricted only to the pathogenesis of different diseases. It also determines the response to some forms of therapy, with two effects described in the literature: (1) the regulation of lymphocyte sensitivity to glucocorticoids by decreasing cellular proliferation of peripheral blood mononuclear cells and (2) the modulation of the response to inhaled short-acting bronchodilators (BD) [23,24]. 

A recent study using CRISPR interference (CRISPRi) tried to assess the molecular mechanisms connected to the genes associated with chronic obstructive pulmonary disease (COPD) and low lung function. After a GWAS analysis searching for genes related to COPD, the experiments were conducted on human-induced pluripotent stem cell (iPSC)–derived lung epithelium. The results of this study show that the knockdown of *RBMS3* enhances the proliferation of cells, which is the basis for later experiments clarifying the exact role of RBMS3 in COPD [30].

## 5. Role of RBMS3 in Carcinogenesis

In 2008, RBMS3 was mentioned in the context of neoplastic processes for the first time [31]. From that time onwards, RBMS3 has significantly grown in popularity and importance as a potential marker and regulator in many different types of cancer. The increasing amount of scientific data provided by researchers has started to unveil the specific mechanisms of RBMS3’s impact on carcinogenesis and metastasis (Table 2).

### 5.1. Bladder Cancer 

The results showed that the downregulation of RBMS3 in bladder cancer was specifically related to a better overall survival (OS), with a higher expression of RBMS3 implicating a poorer prognosis. This was confirmed a few months later by Chen et al. The expression of RBMS3 was also significantly correlated with grade and stages T and M in the TNM scale [32,33].

### 5.2. Gallbladder Carcinoma (GBC) 

The relationship between the expression of RBMS3 and bladder cancer is one of the most recently discussed in the literature. While studying the role of RBMS3 in gallbladder carcinoma, scientists found its downregulation at the mRNA, and protein levels in the tested specimens had an impact on their overall survival. A low expression correlated with a worse OS and acted as an independent negative prognostic factor. Moreover, the overexpression of RBMS3 successfully inhibits growth and promotes the apoptosis of GBC cell lines in in vitro studies. A low expression of RBMS3 also leads to increased angiogenesis, highlighting another process influenced by this protein [34]. 

### 5.3. Prostate Cancer 

Studies on prostate cancer provided evidence of another biological mechanism of the role of RBMS3 in carcinogenesis. *RBMS3-AS3*, a long noncoding RNA (lncRNA), was found to play a significant role as an antitumor factor. LncRNAs are noncoding RNA fragments longer than 200 nucleotides with the ability to bind to different microRNAs (miRNAs) functioning as competing endogenous RNA (ceRNA) [35]. *RBMS3-AS3* binds competitively to miR-4534, increasing the level of its downstream target vasohibin 1 (VASH1), creating the molecular axis *RBMS3-AS3*/miR-4534/VASH1, which may play a pivotal role in prostate cancer development and treatment. *RBMS3**-AS3* is downregulated in prostate cancer, which leads to an upregulation of miR-4534, which decreases the level of VASH1. Experimental upregulation of *RBMS-AS3* led to the inhibition of tumor growth, angiogenesis, and migration by the upregulation of VASH1. VASH1 as a downstream target is also important because it can work as an individual prognostic marker of prostate cancer, and recent studies have shown that its upregulation can inhibit lymphangiogenesis [36]. Another product of the *RBMS3* gene belongs to the group of circular RNAs (circRNAs) containing noncoding RNA with various functions. has_circ_0064644 was the most downregulated circRNA in prostate cancer. Its exact role in prostate cancer progression is yet to be revealed [37].

### 5.4. Epithelial Ovarian Cancer (EOC)

Managing patients with ovarian epithelial cancer is still an exceedingly challenging task for oncologists due to the high rate of relapses caused by chemoresistance. Platinum-based therapy, combined with surgical cytoreduction, is still one of the most effective methods of treatment in EOC. The studies conducted to elucidate the role of RBMS3 in EOC provided data to support the statement that the deletion of the region of chromosome 3 containing the gene for RBMS3 is correlated with a poorer prognosis and acts as an independent prognostic factor for relapse-free survival in this type of cancer. The deletion of *RBMS3* leads to the development of chemoresistance in the patient-derived xenograft (PDX) model and in EOC cell lines. The molecular mechanism underlying these results consists of several elements. First, the loss of *RBMS3* promotes efflux in EOC cells, preventing cytotoxic platinum from getting into the cells. The downregulation of RBMS3 significantly decreases platinum-induced DNA damage and apoptosis, indicating a potential role in restricting DNA damage repair. The lack of RBMS3 activates the Wnt/β-catenin pathway by allowing the strong negative regulator miR-126-5p to downregulate strong Wnt/β-catenin repressors. RBMS3 takes part in the competitive stabilization of many identified repressors, including DKK3, AXIN1, BACH1, and NFAT5 [38]. The *RBMS3* gene was also used in the creation of the tumor-mutation-burden-related signature model. This is a model that uses the total number of replacement and insertion/deletion (indel) mutations per basic group in the exon coding region of the assessed gene in the genome of a tumor cell to predict overall survival in a specific cancer, in this case, ovarian cancer [39]. 

### 5.5. Nasopharyngeal Cancer (NPC) 

Studies conducted on nasopharyngeal cancer introduced RBMS3 as a potential regulator of the cell cycle. Researchers provided evidence of the significant downregulation of RBMS3 in NPC cell lines and postoperational tumor specimens. The ectopic expression of RBMS3 proved to have the ability to inhibit tumor growth and foci formation. As the reason for these abilities, scientists provided a number of molecular mechanisms related to the cell cycle, including apoptosis and microvessel formation. RBMS3 increased the level of p53, which plays a crucial role in promoting the cell cycle from the G1 phase to the S phase. The upregulation of p53 creates a cascade of effects that prevent cells from going further in the cell cycle. An increased expression of p53 increases the expression of p21, which has the ability to suppress the cell cycle by inhibiting the complex cyclin E/CDK2. This complex has an influence on retinoblastoma proteins (RBs), decreasing their phosphorylated inactive form in favor of the unphosphorylated one, which has the ability to stop cells from reaching the next stage of the cell cycle. The overexpression of p53 along with MMP2 and MMP9 may also have an impact on the inhibition of microvessel formation by RBMS3. Changes in the expression of MMP2, MMP9, MMP7, and c-Myc may be explained by the inhibited nuclear translocation of β-catenin. C-Myc is an important downstream target of the Wnt/β-catenin pathway in this case, since its expression correlates with a poorer prognosis, and there is evidence of RBMS3’s abilities to bind to the promoter region of *c-Myc*. The role of RBMS3 in the increased apoptotic activity of NPC was explained with the activation of caspase 9 and PARP by RBMS3 [40,41].

### 5.6. Gastric Cancer (GC) 

All studies concerning the connection between the expression of RBMS3 and gastric cancer provided information about the downregulation of RBMS3 in this type of cancer. RBMS3 was found to have an impact on the secreted frizzled-related protein 1 (SFRP1), playing a significant role in the downregulation of the Wnt/β-catenin pathway by the competitive inhibition of Wnt-frizzled membrane receptor (Fzs) complexes. The low expression of RBMS3 and SFRP1 was found to correlate with a poorer prognosis. The expression of both proteins is statistically related to a poor histological grade and prognosis. The combined expression of RBMS3 and SFP1 acts as an independent prognostic factor in GC. Another downstream target regulated by RBMS3 in GC is the basic helix-loop-helix-PAS transcription factor α (HIF1-A) subunit of the HIF-1 protein, responsible for the induction of VEGF expression in cancer cells. VEGF is a key factor responsible for angiogenesis in tumors. The expression of HIF1-A is increased in GC cells. This, combined with a decreased level of the RBMS3 expression, correlates with a poor histopathological differentiation and a stronger angiogenesis. The overexpression of RBMS3 in GC cells revealed an increased percentage of cells in the G0/1 phase and a lower number of cells in the S phase of the cell cycle, but it had no statistically significant influence on cells in the stage G2/M. Additionally, lower expressions of CDK1, CDK6, E2F1, and MYC were observed, providing evidence of RBMS3’s impact on the cell cycle in GC. RBMS3 also has an impact on circular RNA (circRNA) single-stranded enclosed RNAs, which are common regulators of carcinogenesis. *CircRBMS3* is postulated to be tied with an advanced TNM stage, poor differentiation, larger tumor size, and lymph node metastasis positivity by the regulation of miR-153 and SNAIL1. The overexpression of *circRBMS3* was also shown to be connected to a lower OS. The artificial knockdown of *circRBMS3* results in the inhibition of tumor growth and invasiveness [8,42,43].

### 5.7. Esophageal Squamous Cell Carcinoma (ESCC)

The loss of the 3p fragment of chromosome 3 is one of the most common chromosomal alterations in esophageal squamous cell carcinoma. One of the frequently lost genes is *RBMS3* [44]. The downregulation of RBMS3 significantly correlates with poorer outcomes in patients with ESCC. The ectopic expression of RBMS3 results in tumor growth impairment confirmed by foci formation and tumor xenograft formation tests. Experimental data point to the downregulation of c-Myc and CDK4 as the mechanism mediating RBMS3’s tumor suppressive gene (TSG) abilities. Interestingly, other cell-cycle-related proteins, such as CDK2 or cyclin E or D1, dysregulated in other types of cancer, do not seem to be involved in RBMS3’s role in ESCC. Further studies showed that Rb, the downstream target of CDK2, was also found to be altered by the expression of RBMS3. A decreased level of CDK2 increases the level of inactivated phosphorylated Rb at Ser807/811 and Ser780 [45].

### 5.8. Lung Cancer

Depending on the type of lung cancer, different approaches to the role of RBMS3 were taken, highlighting different aspects of RBMS3’s effect on lung cancer progression. Lung squamous cell carcinoma (LSCC) was characterized by the downregulation of RBMS3 and the upregulation of c-Myc and β-catenin. Oddly enough, there was only a statistically significant correlation of RBMS3’s expression with c-Myc. The combined positive expression of RBMS3 and negative expression of c-Myc act as an independent prognostic factor of shorter OS [46]. As for small-cell lung cancer (SCLC), Xiuwei Li et al. provided evidence of the downregulation of RBMS3 and its upstream miRNA hsa-miR-7-5p by using bioinformatic methods. Hsa-miR-7-5p was previously reported to display tumor-suppressive properties in glioma and glioblastoma by the regulation of the EGFR, PI3K/ATK, Raf/MEK/ERK, and IGF-1R pathways [52,53]. Another type of lung cancer discussed in the context of RBMS3 expression was non-small-cell lung cancer (NSCLC). By using computational methods, scientists identified RBMS3 as a core transcription factor regulating lung-adenocarcinoma-associated genes [54]. Other bioinformatic analyses provided evidence of RBMS3 belonging to the group of genes most negatively correlated with tumorigenesis and being dysregulated in precancer cells. Furthermore, this dysregulation advances through cancer progression [55].

### 5.9. Papillary Thyroid Cancer

The analysis of lncRNA in papillary thyroid cancer revealed that another product of *RBMS3*’s expression, *RBMS3-AS1*, is closely associated with a patient’s shorter OS, broadening the variety of tumors in which RBMS3 has the potential to be a diagnostic marker [47].

### 5.10. Hepatocellular Carcinoma (HCC) 

Studies conducted on hepatocellular carcinoma present RBMS3 in a position of effector instead of regulator. In this case, an upregulated miR-1269 is responsible for altering the expression of *RBMS3* and eight other genes: *AGAP1*, *AGK*, *BMPER*, *BPTF*, *C16orf74*, *DACT1*, *LIX1L*, and *ZNF706* [56]. 

### 5.11. Neuroblastoma

The potential role of RBMS3 in the carcinogenesis of the neuroblastoma was discovered through high-resolution array copy number analyses that showed the presence of homozygous deletion on 3p. However, there are no further studies on this issue [31]. 

### 5.12. Breast Cancer (BC) 

The role of RBMS3 has been most extensively explored in breast cancer among all types of cancer. The expression profile of *RBMS3* at the protein and RNA levels is downregulated. The overexpression of *RBMS3* inhibits the growth, invasion, and migration of BC cells. In vivo experiments conducted in mice also showed an attenuation of tumor growth. As for the clinicopathological characteristics, the downregulation of *RBMS3* correlates with a poor prognosis and a shorter OS. A negative ER status corresponding with the expression of RBMS3 and the combined expression of both these parameters act as independent prognostic factors. The molecular mechanisms underlying these effects include the impact on the Wnt/β-catenin pathway and the cell cycle, confirmed by the inhibited expression of β-catenin, c-Myc, and cyclin D1 in RBMS3 expressing cancer cells [7,48]. Another point of regulation lies in the lncRNA (long noncoding RNA) maternally expressed gene 3 (MEG3)-miR-141-3p-RBMS3 axis. LncRNA encoded by *MEG3* was found to have tumor-suppressive abilities in different types of tumors, including glioma, gastric cancer, and melanoma. MiR-141-3p is a microRNA (miRNA) belonging to the miR-200 family dysregulated in many tumors. An overexpression of miR-141-3p was found in bladder cancer and esophageal squamous cell carcinoma. A low expression of MEG3 upregulates miR-141-3p, which anterogradely downregulates RBMS3 in BC. MEG3 is a tumor-suppressive gene regulating AKT and NF-κB signaling pathways, inducing apoptosis through its impact on Bcl-2 and C casp-3 and p53 signaling. MiR-141-3p is a miRNA whose role depends on the type of tumor, with capabilities ranging from tumor-suppressive abilities to overexpression correlated with poor prognosis and chemoresistance [49,50,51]. Moreover, a recent study showed that the *RBMS3* gene expression in the tumor-associated stromal cells of breast tumor was gradually downregulated among grade I, II, and III of breast cancer. The downregulation of this gene was also correlated with worse clinical outcome and poorer survival prognosis [57].

## 6. Epithelial–Mesenchymal Transition and Role of RBMS3 in This Process

Epithelial–mesenchymal transition (EMT) is a biological process that allows epithelial cells to switch their phenotype to quasi-mesenchymal [58,59,60]. EMT causes epithelial cells to lose characteristic features, such as tight cell–cell junctions [61] and cell polarity [59], and acquire mesenchymal properties instead [62]. This is first observed during embryogenesis, in gastrulation or tissue morphogenesis [63]. Furthermore, the process plays a crucial role in wound healing, fibrosis, and tumor progression [60,61,62,63,64,65,66]. The reverse process is called MET, from mesenchymal–epithelial transition, and it occurs when the mesenchymal cells acquire epithelial characteristics [67].

Typically, epithelial cells appear as cells attached to basal lamina, with tight cell–cell junctions and apical–basal polarity [68]. When it comes to EMT, epithelial cells lose these properties and the ability of the expression of E-cadherin—a molecule that is essential to maintaining the epithelial phenotype [58]. The loss of E-cadherin is considered to be a hallmark of EMT along with the acquisition of the expression of vimentin of N-cadherin [68]. During EMT, the epithelial cells, which have a typical cobblestone morphology, transform into quasi-mesenchymal cells, which have a rather fibroblastic-like phenotype [69]. This transition allows cells to acquire a migratory phenotype and become more invasive [70]. These changes in phenotype require rearrangements of the cytoskeleton and the cell metabolism [71]. Due to the acquisition of these properties, EMT plays a significant role in tumor progression, metastasis, and malignancy [58,71]. 

Three types of EMT processes can be distinguished. Type 1 describes an EMT that occurs in the development of tissues. The EMT subtype that occurs in fibrosis and wound healing is type 2, with type 3 being observed in cancer cells [72]. Although, historically, EMT was discovered by developmental biologists [63], modern studies focus on the link between EMT and cancer [73]. Recent observations suggest that EMT is also involved in the therapeutic resistance of various tumors [67,74,75]. 

EMT is a process that is strictly determined by genetic mechanisms. Several transcription factors involved in this phenotype change have been discovered [76]. Some well-described EMT-TFs (epithelial–mesenchymal transition transcription factors) are SNAIL1, SNAIL2, TWIST1, and ZEB1. However, the list of EMT-TFs is way longer, and there are many more transcription factors involved in EMT, for instance, FOX- or SOX-TF [76]. The crucial signaling pathways of EMT are Wnt and TGF-β, but other pathways, such as Notch or Hedgehog, are also involved [68]. Some of the descriptions of the molecular mechanisms seem to be quite preposterous; thus, there is still a lot of speculation and uncertainty surrounding the topic.

As it has already been mentioned, EMT plays a major role in cancer progression. EMT allows cancer cells to become more mesenchymal-like. EMT is probably responsible for the creation of circulating tumor cells (CTCs), which are strictly connected to the ability to metastasize [77]. CTCs are an element of the invasion-metastatic cascade, and EMT is believed to be involved it this type of tumor progression [58]. It is worth noticing that the reverse process, mesenchymal–epithelial transition (MET), is also important for the ability of cancer cells to metastasize [67,78,79]. EMT is considered to be a relevant process in the development of cancer steam cells (CSCs). Therefore, it could be responsible for therapeutic resistance [58,78]. 

With a better understanding of EMT’s complexity and its importance and vital role in cancer progression, invasion, and the development of metastases and CSCs comes the necessity to find and describe the key regulators of this process. RBMS3 is a novel potential regulator of EMT, with an increasing amount of data trying to unveil its molecular role in this process. Figure 2 and Table 3 present the currently proposed mechanisms of the impact of RBMS3 on the EMT process. 

The Wnt/β-catenin signaling pathway is a critical molecular mechanism regulating the EMT process. Downstream targets of Wnt include, among others, Twist, Snail, and MMP7 genes facilitating EMT [9]. In several of the previously discussed types of cancer, the Wnt/β-catenin pathway was inhibited by RBMS3’s expression. The expression of c-Myc, another downstream target of the Wnt/β-catenin pathway, was also investigated and found statistically correlated with RBMS3’s expression. 

Several studies focus solely on RBMS3’s interaction with the EMT machinery. While studying breast cancer in 2019, Zhu L et al. identified a regulatory axis consisting of RBMS3, TWIST1, and matrix metalloproteinase 2 (MMP2), responsible for the migration and invasion of the tumor. The expression of RBMS3 downregulated the expression of TWIST1, one of the key factors of EMT, and consecutively, its downstream target MMP2, leading to EMT impairment and invasion and migration inhibition [9]. Another study conducted on breast cancer provided interesting data stating that the expression of RBMS3 is a required factor for EMT induction in immortalized mammary epithelial cell lines. In the triple negative breast cancer (TNBC) model, RBMS3 was essential for maintaining the mesenchymal phenotype, invasiveness, and migration ability. In vivo experiments showed the loss of RBMS3 to impair the growth of the tumor and its ability to create metastasis. As the potential molecular basis of this process, the authors indicated RBMS3’s ability to influence expression and stabilize *PRRX1* mRNA, a transcription factor regulating EMT [80]. Research conducted on gastric cancer by Zhao seems to be coherent with Zhu’s results, showing an increased expression of E-cadherin and a decreased expression of N-cadherin and β-catenin in RBMS3-overexpressing gastric cancer cells. Moreover, an increased expression of RBMS3 significantly decreased the invasive abilities of cells [81]. 

Taking into consideration all the information contained in this chapter, there is convincing evidence of RBMS3 being one of the regulators involved in the EMT process, even though the exact mechanism of this regulation requires further investigation and may differ depending on the molecular subtype of cancer. 

**Table 3 ijms-23-10875-t003:** Proposed molecular mechanisms of RBMS3’s impact on EMT.

Type of Cancer	Currently Proposed Mechanisms of RBMS3’ Impact on EMT
Breast cancer	The expression of RBMS3 downregulates the expression of TWIST1 and, consecutively, its downstream target MMP2, leading to EMT impairment [9] Loss of RBMS3 impairs the growth of the tumor and its ability to create metastasis by influencing expression and stabilizing *PRRX1* mRNA, a transcription factor regulating EMT [80]
Gastric cancer	Increased expression of E-cadherin and decreased expression of N-cadherin and β-catenin in RBMS3-overexpressing cancer cells [81]

## 7. Conclusions

All the information provided in this review depicts *RBMS3* as a functionally versatile gene that uses its main and multiple alternative products of expression to play a significant role in embryonic development and the pathogenesis of many different diseases, especially the induction and progression of cancers. The main ways in which RBMS3 impacts cells are the regulation of the expression or stabilization of miRNA, inhibiting Wnt/β-catenin signaling pathway and other EMT-related transcription factors. These molecular characteristics make RBMS3 a promising biomarker of OS and a prognostic factor in neoplastic processes, where statistical data support this statement for many different types of cancer. Another potential use of RBMS3 is as a target for anticancer drugs, thanks to its function as TSG and its proven ability to suppress cancer migration and invasive abilities. Artificially increased expression of RBMS3 utilizing genome editing techniques may potentially improve the outcome of standard therapies in many types of cancers. Increased expression of RBMS3 may prevent the creation of micrometastases that are too small to be picked up in diagnostic imaging and may lead to relapse of tumor.

There are some limitations to targeting RBMS3 mainly concerning the lack of a deep understanding of molecular mechanisms that are responsible for RBMS3 tumor suppressive abilities and the regulation of this properties. Additionally, currently, there are not enough data concerning the role of RBMS3 expression in different types of healthy human tissues and the consequences of RBMS3 level alteration. However, the year-on-year increasing amount of data and the incoherencies of some of the results indicate that the molecular role of RBMS3, especially in the regulation of cancer development, is a good subject for further research that may lead to the development of novel diagnostic and therapeutic strategies that will improve the outcome of patients with neoplastic diseases.

## Figures and Tables

**Figure 1 ijms-23-10875-f001:**

Workflow literature review.

**Figure 2 ijms-23-10875-f002:**
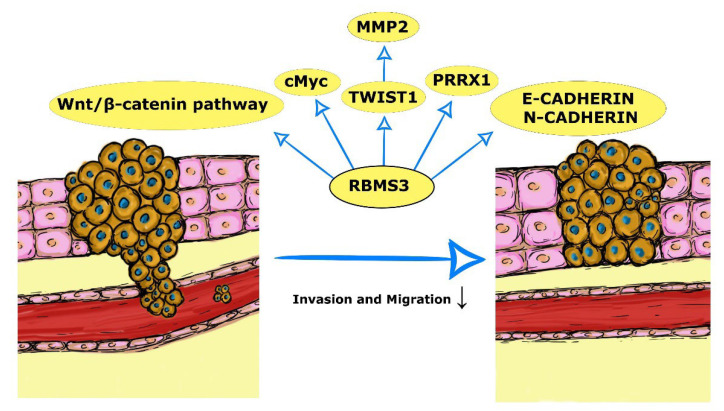
Mechanisms of the impact of RBMS3 on the EMT process.

**Table 1 ijms-23-10875-t001:** Single-nucleotide polymorphisms of the *RBMS3* gene and their correlation with pathological processes.

RBMS3-Related Processes	Identified SNP	References
Bone-mineral-density-related disorders	rs6549904 rs7640046 rs17024608	[15]
Osteonecrosis of the jaw (ONJ)	rs17024608	[17]
Exfoliation glaucoma	rs12490863	[18]
Exfoliation syndrome	rs12490863	[19]
Primary Sjögren’s syndrome	rs13079920 rs13072846	[20]
Systemic sclerosis	rs1449292	[21]
Periodontal disease	rs17718700	[22]
Lymphocyte glucocorticoid sensitivity	rs6549965	[23]
Short-acting bronchodilator response	rs1266115 rs150703870	[24]

**Table 2 ijms-23-10875-t002:** Role of RBMS3 in carcinogenesis.

Tumor Type	Correlation with High or Low Expression of RBMS3	Mechanism of Action	References
Bladder cancer	High expression correlates with poorer prognosis.	Further research is needed.	[32,33]
Gallbladder carcinoma	Low expression correlates with shorter OS. High expression inhibits growth and promotes apoptosis in vitro.	Further research is needed.	[34]
Prostate cancer	Upregulation of *RBMS-AS3* correlates with faster tumor growth, angiogenesis, and migration.	*RBMS-AS3*/miR-4534/VASH1 axis.	[35,36,37]
Ovarian epithelial cancer	Loss of *RBMS3* gene is correlated with poorer prognosis. Deletion of *RBMS3* promotes efflux and induces chemoresistance.	RBMS3 promotes efflux. Lack of RBMS3 activates the Wnt/β-catenin pathway.	[38,39]
Nasopharyngeal cancer	Ectopic expression inhibits tumor growth and foci formation.	RBMS3 increases the level of p53, and thus p21 and MMP2 and MMP9. c-Myc/Wnt/β-catenin axis.	[40,41]
Gastric cancer	Low expression correlates with poorer prognosis, poor histological grade, and angiogenesis.	Wnt/β-catenin pathway. Low expression of RBMS3 induces overexpression of HIF1-A.	[8,42,43]
Esophageal squamous cell carcinoma	Low expression correlates with poorer prognosis. Ectopic expression inhibits tumor growth.	RBMS3 induces downregulation of c-Myc and CDK4.	[44,45]
Lung cancer	Low expression correlates with worse OS.	Downregulation of RBMS3 and upregulation of c-Myc and β-catenin.	[46]
Papillary thyroid cancer	High expression of *RBMS3-AS1* correlates with shorter OS.	Further research is needed.	[47]
Breast cancer	High expression inhibits tumor growth, invasion, and migration. Low expression correlates with poorer prognosis and shorter OS. Levels of expression of ER and RBMS3 are correlated.	Wnt/β-catenin axis. MEG3-miR-141-3p-RBMS3 axis.	[7,48,49,50,51]

## Data Availability

Not applicable.

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
