# Peer review of "Role of RBMS3 Novel Potential Regulator of the EMT Phenomenon in Physiological and Pathological Processes"

_ijms, 2022, doi:10.3390/ijms231810875_

Round 1

Reviewer 1 Report (Previous Reviewer 1)

The objective of this review article is to report the role of RNA binding protein, RNA-binding motif single-stranded-interacting protein 3 (RBMS3) in physiological and pathological processes with emphasis on carcinogenesis. The authors have discussed the role of RBMS3 in physiology, pathological processes (non-cancerous) and in different types of cancers. It is interesting review article. The authors have addressed the concerns and the manuscript could be considered for publication.

Author Response

In accordance with the comments and suggestions received from reviewers, our manuscript (ijms-1927966) has been revised.

Detailed responses to Reviewer’s comments are listed below:

The objective of this review article is to report the role of RNA binding protein, RNA-binding motif single-stranded-interacting protein 3 (RBMS3) in physiological and pathological processes with emphasis on carcinogenesis. The authors have discussed the role of RBMS3 in physiology, pathological processes (non-cancerous) and in different types of cancers. It is interesting review article. The authors have addressed the concerns and the manuscript could be considered for publication.

Response: Thank you very much for your evaluation and recommendations. We are very thankful for positive revision of our manuscript.

Reviewer 2 Report (Previous Reviewer 3)

The authors have discussed about the potential role of the role of RBMS3 in physiological and pathological process with particular emphasis on cancer. The manuscript is well compiled and authors need to address few points as indicated below:

1. The novelty of the article should be clearly highlighted as few reviews have already been published on this topic.

2. More important references from last few years should be added if available to improve visibility and quality of current work.

3. The various limitations associated in targeting RBMS3 should be discussed.

4. The authors should include additional tables to convey the key message of the article.

5. The authors should provide their own justification and relevance of the study. This will help the readers to understand the importance of the paper.

6. Typographical errors were found throughout the text and should be corrected.

Author Response

In accordance with the comments and suggestions received from reviewers, our manuscript (ijms-1927966) has been revised.

Detailed responses to Reviewer’s comments are listed below:

The authors have discussed about the potential role of the role of RBMS3 in physiological and pathological process with particular emphasis on cancer. The manuscript is well compiled and authors need to address few points as indicated below:

  1. The novelty of the article should be clearly highlighted as few reviews have already been published on this topic.

Response: Thank you very much for your evaluation and recommendations, we focused on the role of RMBS3 in the Epithelial-Mesenchymal Transition, especially in cancerous processes. According to our best knowledge so far there have not been published any systematic review of the role of RBMS3 in physiological and pathological processes. We made sure to highlight the novelty of the article in the Introduction section by highlighting that “There is still need for a summary of the role of RBMS3 in physiology and pathology…”

  1. More important references from last few years should be added if available to improve visibility and quality of current work.

Response: Thank you very much for your suggestion, we added newest references concerning role of RBMS3 expression in stroma of breast cancer (reference number 62) in order to increase quality of our manuscript.

  1. The various limitations associated in targeting RBMS3 should be discussed.

Response: Thank you very much for your suggestion we discussed the limitations associated with RBMS3 targeting  in Conclusion section of our manuscript: “ There are some limitations to targeting RBMS3 mainly concerning lack of in depth understanding of molecular mechanisms that are responsible for RBMS3 tumour suppressive abilities and regulation of this properties. There is also currently not enough data concerning role of RBMS3 expression in different types of healthy human tissues and consequences of RBMS3 level alteration.”

  1. The authors should include additional tables to convey the key message of the article.

Response: Thank you very much for your evaluation and recommendations we added additional table (Table 3) that summarize proposed mechanisms of RBMS3 role in EMT process to convey the key message of our manuscript.

  1. The authors should provide their own justification and relevance of the study. This will help the readers to understand the importance of the paper.

Response: Thank you very much for your suggestion. In order to clarify importance of the paper we added additional justification in the conclusion section: “ The year-on-year increasing amount of data and the incoherencies of some of the results indicate that the molecular role of RBMS3, especially in the regulation of cancer development, is a good subject for further research that may lead to development of novel diagnostic and therapeutic strategies that will improve outcome of the patients with neoplastic disease.”

  1. Typographical errors were found throughout the text and should be corrected.

Response: Thank you very much for your evaluation, we made sure to correct typographical errors found in text.

This manuscript is a resubmission of an earlier submission. The following is a list of the peer review reports and author responses from that submission.

Round 1

Reviewer 1 Report

The objective of this review article is to report the role of RNA binding protein, RNA-binding motif single-stranded-interacting protein 3 (RBMS3) in physiological and pathological processes with emphasis on carcinogenesis. The authors have discussed the role of RBMS3 in physiology, pathological processes (non-cancerous) and in different types of cancers. It is interesting review article but the manuscript could not be considered for publication in the current form for following reasons.

Major concerns:

1) The manuscript is poorly written and needs extensive paraphrasing of the sentences. The authors intended message could have been conveyed with better sentence structure. The authors are recommended to use language editing services either from MDPI or other accepted/approved English editing service providers.

2) Could authors please include sub-titles to denote type of the cancer in the "role of RBMS3 in carcinogenesis" section.

3) Could authors please carefully review and rephrase sentence structures/choice of words. Example: Please paraphrase the sentence "What is more authors have also provided evidence...." on Ln # 62 of page #2; "determinate" on Ln # 133 of page # 4; "From that time onward has significantly grown in popularity" on Ln # 145 of page # 4; and "researches" on Ln # 222 of page # 7.

4) Could authors please carefully proof read the manuscript and correct the grammatical and syntax errors. 

Author Response

In accordance with the comments and suggestions received from reviewers, our manuscript (ijms-1856638) has been revised.

Detailed responses to Reviewer’s comments are listed below:

The objective of this review article is to report the role of RNA binding protein, RNA-binding motif single-stranded-interacting protein 3 (RBMS3) in physiological and pathological processes with emphasis on carcinogenesis. The authors have discussed the role of RBMS3 in physiology, pathological processes (non-cancerous) and in different types of cancers. It is interesting review article but the manuscript could not be considered for publication in the current form for following reasons.

Major concerns:

1) The manuscript is poorly written and needs extensive paraphrasing of the sentences. The authors intended message could have been conveyed with better sentence structure. The authors are recommended to use language editing services either from MDPI or other accepted/approved English editing service providers.

Response: Thank you very much for you evaluation and recommendations. It is very important for us to preserve the highest possible standards so we decided to use English editing service.

2) Could authors please include sub-titles to denote type of the cancer in the "role of RBMS3 in carcinogenesis" section.

Response: Thank you very much for your suggestions we added sub-titles in the 5th paragraph

3) Could authors please carefully review and rephrase sentence structures/choice of words. Example: Please paraphrase the sentence "What is more authors have also provided evidence...." on Ln # 62 of page #2; "determinate" on Ln # 133 of page # 4; "From that time onward has significantly grown in popularity" on Ln # 145 of page # 4; and "researches" on Ln # 222 of page # 7.

Response: Thank you very much for your suggestions in order to increase readability of our work we review and rephrase sentence structures and emphasis on choice of words.

4) Could authors please carefully proof read the manuscript and correct the grammatical and syntax errors. 

Response: Thank you very much for you evaluation and recommendations. It is very important for us to preserve the highest possible standards so we decided to use English editing service.

Reviewer 2 Report

The authors proposed a review article on the role of RBMS3 on physiological and pathological processes with particular reference to EMT process. The structure of the manuscript needs improvements as well as English. Some parts are not well described. Significant improvements are needed. See the comment below:

1) A Figure with the workflow used for the selection of paper should be added;

2) The manuscript needs some English editing performed by a native speaker (“was” in line 50; grammar in line 94; etc.);

In chapter 4 the authors just mentioned the association between RBMS3 with non-cancerous pathologies. However, the role of RBMS3 or the molecular mechanisms responsible for the development of these diseases were not properly described;

It is not clear what is the main topic of the manuscript. It is described in a very confusing manner and each chapter is not properly linked with the others. The structure of the manuscript needs significant improvements;

Chapter 6 should be shortened as redundant information on EMT are provided. 

Author Response

In accordance with the comments and suggestions received from reviewers, our manuscript (ijms-1856638) has been revised.

Detailed responses to Reviewer’s comments are listed below:

The authors proposed a review article on the role of RBMS3 on physiological and pathological processes with particular reference to EMT process. The structure of the manuscript needs improvements as well as English. Some parts are not well described. Significant improvements are needed. See the comment below:

  • A Figure with the workflow used for the selection of paper should be added

Response: Thank you very much for suggestion we added workflow diagram to method section of our article (Figure1).

  • The manuscript needs some English editing performed by a native speaker (“was” in line 50; grammar in line 94; etc.);

Response: Thank you very much for your evaluation and recommendations. It is very important for us to preserve the highest possible standards so we decided to use English editing service.

  • In chapter 4 the authors just mentioned the association between RBMS3 with non-cancerous pathologies. However, the role of RBMS3 or the molecular mechanisms responsible for the development of these diseases were not properly described;

Response: Thank you very much for your advice. In order to provide most complete picture of RBMS3 role in organism we wanted to highlight presence of reports connecting expression of RBMS3 with various non-cancerous pathologies. Currently not very much is known about molecular mechanisms lying  underneath the role of RBMS3 in these pathologies and further studies are required but we vividly inspect this part and edited it to make sure it is properly described.

  • It is not clear what is the main topic of the manuscript. It is described in a very confusing manner and each chapter is not properly linked with the others. The structure of the manuscript needs significant improvements;

Response: Thank you very much for suggestion we made sure to increase the clarity of our article by improving its body. 

  • Chapter 6 should be shortened as redundant information on EMT are provided. 

Response: Thank you very much for your evaluation and recommendations it is very important for us to allow everyone not familiar with topic of this article to properly understand processes we describe as we provided broader range of information about EMT phenomenon.

Reviewer 3 Report

The authors have discussed about the potential role of RBMS3 in physiological and pathological process with particular emphasis on carcinogenesis. Specific points that the authors need to address are as follows:

1. The novelty of the article should be clearly highlighted as some reviews have already been published on this topic.

2. More important references from last few years should be added if available to improve visibility and quality of current work.

3. The various factors that can determine the tumor promoter/or suppressor functions of RBMS3 should also be discussed.

4. The strategies for modulating RBMS3 expression to suppress tumor growth and metastasis should be discussed.

5. The authors should provide their own justification and relevance of the study. This will help the readers to understand the importance of the paper.

6. Several typographical errors were found throughout the manuscript and should be corrected.

Author Response

In accordance with the comments and suggestions received from reviewers, our manuscript (ijms-1856638) has been revised.

Detailed responses to Reviewer’s comments are listed below:

The authors have discussed about the potential role of RBMS3 in physiological and pathological process with particular emphasis on carcinogenesis. Specific points that the authors need to address are as follows:

1.The novelty of the article should be clearly highlighted as some reviews have already been published on this topic.

 Response: Thank you very much for your evaluation and recommendations we made sure to clearly highlight the novelty of our work in the first paragraph of the article by  adding an additional sentences in introduction paragraph: “RBMS3 ability to suppress growth and progression of different types of cancers makes it potential target for development of novel anti-cancer therapies. There is still need for a summary of the role of RBMS3 in physiology and pathology that would provide a synthetic evaluation of the information available about it”

  1. More important references from last few years should be added if available to improve visibility and quality of current work.

Response: Thank you very much for your suggestions. Working on this article we have chosen carefully most recent and important reports about RBMS3 to provide highest quality of our work.

  1. The various factors that can determine the tumor promoter/or suppressor functions of RBMS3 should also be discussed.

Response: Thank you very much for your advice. Outside of miR-1269 described in part 5.10 of the article about hepatocellular cancer according to our knowledge there are no other hypothesis describing factors that can determine tumor promoter/suppressor function of RBMS3. In our work we presented in details the most recent reports about the RBMS3.

  1. The strategies for modulating RBMS3 expression to suppress tumor growth and metastasis should be discussed.

Response: Thank you very much for your recommendations. As molecular mechanisms underlying RBMS3 tumor suppressing abilities require further investigation. According to our knowledge there are no reports describing strategies for modulation of RBMS3 expression. Hypothetical use of RBMS3 protein in anti-cancer therapies was discussed in the concluding part of our work: Artificially increased expression of RBMS3 utilizing genome editing techniques may potentially improve outcome of standard therapies in many types of cancers. Increased ex-pression of RBMS3 may prevent creation of micrometastases that are too small to be picked up in diagnostic imaging and may lead to relapse of tumour. “

  1. The authors should provide their own justification and relevance of the study. This will help the readers to understand the importance of the paper.

Response: Thank you very much for your suggestion.  We provided additional justification of study relevance in first paragraph of our work as RBMS3 ability to suppress growth and progression of many types of cancer may be very interesting direction for novel anti-cancer therapies development: “RBMS3 ability to suppress growth and progression of different types of cancers makes it potential target for development of novel anti-cancer therapies.”

  1. Several typographical errors were found throughout the manuscript and should be corrected.

Response Thank you very much for you evaluation and recommendations. It is very important for us to preserve the highest possible standards so we decided to use English editing service.

Round 2

Reviewer 3 Report

The authors have addressed all my concerns.